# The Effect of High-Intensity Interval Training Type on Body Fat Percentage, Fat and Fat-Free Mass: A Systematic Review and Meta-Analysis of Randomized Clinical Trials

**DOI:** 10.3390/jcm12062291

**Published:** 2023-03-15

**Authors:** Fatemeh Khodadadi, Reza Bagheri, Raoof Negaresh, Sajjad Moradi, Michael Nordvall, Donny M. Camera, Alexei Wong, Katsuhiko Suzuki

**Affiliations:** 1Department of Exercise Physiology, Ferdowsi University of Mashhad, Mashhad 91779-48974, Iran; 2Department of Exercise Physiology, University of Isfahan, Isfahan 81746-73441, Iran; 3Department of Physical Education & Sport Sciences, Tarbiat Modares University, Tehran 14117-13116, Iran; 4Nutritional Sciences Department, School of Nutritional Sciences and Food Technology, Kermanshah University of Medical Sciences, Kermanshah 67158-47141, Iran; 5Department of Health and Human Performance, Marymount University, Arlington, VA 22207, USA; 6Department of Health and Biostatistics, Swinburne University, Melbourne, VIC 3122, Australia; 7Faculty of Sport Sciences, Waseda University, 2-579-15 Mikajima, Tokorozawa 359-1192, Japan

**Keywords:** body composition, exercise, health, physical activity

## Abstract

This systematic review and meta-analysis of randomized controlled trials (RCTs) compared body compositional changes, including fat mass (FM), body fat percentage (BF%), and fat-free mass (FFM), between different types of high-intensity interval training (HIIT) (cycling vs. overground running vs. treadmill running) as well as to a control (i.e., no exercise) condition. Meta-analyses were carried out using a random-effects model. The I^2^ index was used to assess the heterogeneity of RCTs. Thirty-six RCTs lasting between 3 to 15 weeks were included in the current systematic review and meta-analysis. RCTs that examined the effect of HIIT type on FM, BF%, and FFM were sourced from online databases including PubMed, Scopus, Web of Science, and Google Scholar up to 21 June 2022. HIIT (all modalities combined) induced a significant reduction in FM (weighted mean difference [WMD]: −1.86 kg, 95% CI: −2.55 to −1.18, *p =* 0.001) despite a medium between-study heterogeneity (I^2^ = 63.3, *p =* 0.001). Subgroup analyses revealed cycling and overground running reduced FM (WMD: −1.72 kg, 95% CI: −2.41 to −1.30, *p =* 0.001 and WMD: −4.25 kg, 95% CI: −5.90 to −2.61, *p =* 0.001, respectively); however, there was no change with treadmill running (WMD: −1.10 kg, 95% CI: −2.82 to 0.62, *p =* 0.210). There was a significant reduction in BF% with HIIT (all modalities combined) compared to control (WMD: −1.53%, 95% CI: −2.13, −0.92, *p =* 0.001). All forms of HIIT also decreased BF%; however, overground running induced the largest overall effect (WMD: −2.80%, 95% CI: −3.89 to −1.71, *p =* 0.001). All types of HIIT combined also induced an overall significant improvement in FFM (WMD: 0.51 kg, 95% CI: 0.06 to 0.95, *p =* 0.025); however, only cycling interventions resulted in a significant increase in FFM compared to other exercise modalities (WMD: 0.63 kg, 95% CI: 0.17 to 1.09, *p =* 0.007). Additional subgroup analyses suggest that training for more than 8 weeks, at least 3 sessions per week, with work intervals less than 60 s duration and separated by ≤90 s active recovery are more effective for eliciting favorable body composition changes. Results from this meta-analysis demonstrate favorable body composition outcomes following HIIT (all modalities combined) with overall reductions in BF% and FM and improved FFM observed. Overall, cycling-based HIIT may confer the greatest effects on body composition due to its ability to reduce BF% and FM while increasing FFM.

## 1. Introduction

High-intensity interval training (HIIT) involves repeated bouts of high-intensity work (exercise) performed at or near maximal oxygen uptake (VO_2max_) or above lactate threshold interspersed with low-intensity exercise or periods of rest [1,2,3]. Conceivably coined for the first time in 1912 by Hannes Kolehmainen, the 10,000 m Olympic champion runner [4], HIIT received little attention from athletes until the middle of the 20th century. Gradually since that time, HIIT has become increasingly popular within training regimens for athletes and coaches across sporting disciplines, both for amateurs and professionals alike [4]. Yet research into the effectiveness of HIIT as a viable means to improve fitness began much later when sports scientists compared physiological outcomes to single-component exercises (e.g., endurance exercise alone) [5,6,7]. Unsurprisingly, earlier studies with athletes revealed HIIT promoted significant physiological and performance improvements in submaximal heart rate, VO_2max_, repeated sprint ability, and running/jumping performance [6,8]. More recently, attention has turned to other population cohorts, such as non-athletes, sedentary individuals, and those with chronic diseases such as obesity and cardiovascular disease, to assess whether HIIT can similarly promote beneficial body composition changes compared to performance benefits [9,10]. In this regard, HIIT can incorporate a wide range of exercise modalities (including, but not limited to, cycling, overground running, or treadmill running), intensity, recovery periods (active vs. passive), volume, repetitions, and sets that can be modified to accommodate a variety of populations [10]. Moreover, HIIT provides a time-efficient strategy to meet minimum activity levels [11] and may promote overcoming the often-cited lack of time barrier to initiating an exercise program. Therefore, incorporating HIIT into the exercise training programs of individuals across healthy and clinical cohorts represents a practical and feasible strategy to improve health and fitness. Despite the fact that HIIT has been endorsed by numerous health and fitness professionals [5] as well as the lay media [12,13,14,15,16] as a practical and time-efficient intervention for body composition improvements, meta-analytic investigations on this topic have produced conflicting results. One such meta-analysis found that low-volume HIIT (≤500 metabolic equivalent minutes per week [MET-min/week]) did not improve both lean and fat mass (FM) compared to continuous endurance training and non-exercise controls [17]. However, the authors did not compare different modes of HIIT (cycling, overground running, or treadmill running), which is relevant as body composition changes may be affected by varied movement and muscle activation patterns. On the other hand, a meta-analysis by Wu et al. found that HIIT involving a combination of cycling and whole-body circuit training led to significant improvements in body fat percentage (BF%) and lean mass compared to continuous endurance training in older adults [18]. Moreover, Wewege et al. reported that HIIT incorporating a cycle ergometer and treadmill significantly reduced FM in overweight and obese adults [19]. Based on these inconsistent findings, there is no clear consensus regarding the utility of which type of HIIT induces the greatest improvements in body composition. Therefore, the present systematic review and meta-analysis of randomized controlled trials (RCTs) were conducted to compare the effects of HIIT type, namely cycling, overground running, or treadmill running, on changes in body fat (both FM and BF%) and fat-free mass (FFM). Body compositional changes induced with these different types of HIIT were also compared to a control/no exercise condition. Because overground running causes higher muscular activation compared to other exercise modes [20], we hypothesize that HIIT using overground running may result in improved body composition compared to other modalities. A greater understanding of the type of HIIT that may promote the most beneficial body compositional changes is important to provide robust evidence for practitioners and coaches for helping develop effective and practical HIIT programs.

## 2. Methods

The present study is based on the PRISMA protocol for reporting systematic reviews and meta-analyses [21] and has been registered with PROSPERO (CRD42022323444).

### 2.1. Data Sources and Search Strategies

A comprehensive database search was performed by FK utilizing PubMed, Scopus, Web of Science, and Google Scholar up to 21 June 2022. The following keywords were used in the search strategy: (“High-intensity interval training” or “High-intensity intermittent exercise” or “Aerobic interval training” or “HIIT”) and (“Body composition” or “Fat free mass” or “Skeletal muscle” or “Fat mass” or “Fat percentage”). No language or date of publication restrictions were applied, nor were unpublished investigations considered. Reference lists of relevant studies were manually screened to avoid oversight of eligible investigations not captured in database searches.

### 2.2. Study Selection and Inclusion/Exclusion Criteria

Studies meeting the following eligibility criteria were analyzed for inclusion: (1) RCTs, (2) adult population cohorts (≥18 years), (3) trials reporting mean (SD) alterations to body composition (e.g., subcutaneous skinfold caliper, air displacement plethysmography [ADP], Bioelectrical Impedance Analysis [BIA], and dual-energy X-ray absorptiometry [DEXA]) for both intervention and control (i.e., no exercise intervention) groups or that presented adequate information for calculation of overall effect size, (4) RCTs with a minimum two week (14-day) intervention duration, and (5) trials reporting outcomes in multiple manuscripts from the same data set (e.g., study participants); the original or most complete datasets were included for analysis. It should be noted that RCTs utilizing multiple intervention groups (i.e., more than one HIIT group with different intensities and/or durations) were considered different outcomes, and data were pooled separately in these cases. Investigations and clinical trials that utilized animal models or were performed in children and teenagers (years < 18), were review and/or observational in nature, lacked a control group, or involved population cohorts with a known disease were excluded from the meta-analysis. The acronym PICOS (Population, Intervention, Comparison, Outcomes, and Study Design) was used to develop a focused question and establish inclusion and exclusion criteria for this overview. The PICOS criteria are listed in Table 1.

### 2.3. Data Extraction

Data extraction from each eligible full-text study followed a similar format that included the first author’s name, publication year, sample size (control and intervention groups), participant characteristics (mean age and body mass index [BMI]), duration of intervention, type of HIIT, and mean (SD) percentage and mass (in kg) changes in fat and FFM for both intervention and control groups. In a couple of cases when published data was unclear or ambiguous (i.e., unclear from the publication’s statistical methods whether standard errors, 95% confidence intervals, or SD were used), attempts were made to contact corresponding authors for clarification. Data reported from RCTs using units other than the International System of Units (SI) were converted as applicable.

### 2.4. Risk of Bias Assessment

The Cochrane quality assessment tool was used to evaluate the quality of RCTs [22] in seven domains, including random sequence generation, allocation concealment, reporting bias, performance bias, detection bias, attrition bias, and other sources of bias. Any domain was scored a “high risk” if a study included methodological defects that may have affected outcomes, “low risk” if it did not, and “unclear risk” if the information provided was not sufficient to determine the impact. The overall risk of bias for a randomized controlled trial (RCT) was considered 1: high-quality if four or more domains had “low risk”, 2: moderate-quality if two or three domains had “low risk”, and 3: low-quality if one or less domains had “low risk” [18]. The risk of bias assessment was undertaken independently by two reviewers.

### 2.5. Statistical Analysis

Data compiled for meta-analysis were analyzed using the Stata software, version 14 (StataCorp, College Station, TX, USA), with significant differences considered at *p* ≤ 0.05. The mean change (±SD) of relevant outcomes was used to estimate the overall effect size. When mean changes were not reported (e.g., only body FM in kg pre-and post-intervention was provided), such alterations in body composition (BF% and FFM) were calculated. Standard errors (SEs), 95% confidence intervals (CIs), and interquartile ranges (IQRs) were converted to SDs for subsequent analysis of mean changes in body composition outcomes using the method of Hozo et al. [23]. If the outcome measures were only reported in figures, we used software (GetData Graph Digitizer) to estimate the value. The SD change was calculated based on the following formula: $$SD_{Change}=\sqrt{((SD_{Baseline}{)}^{2}+{\left(SD_{Final}\right)}^{2})-\left(2\times 0.8\times SD_{Baseline}\times SD_{Final}\right)}$$

A random-effects model was applied to account for between-study variations to obtain overall effect sizes. Heterogeneity among RCTs was determined by the I^2^ statistic and Cochrane’s Q test. An I^2^ value > 50% or *p* < 0.05 for the Q-test was considered significant between-study heterogeneity. To find probable sources of heterogeneity, subgroup analyses were performed according to predefined variables, including the type of intervention (cycling vs. overground running vs. treadmill running), duration of the intervention (>8 vs. ≤8 weeks), participant sex (female vs. male) and BMI (<25 vs. 25–30 vs. >30 kg·m^−2^). Sensitivity analysis was conducted by removing RCTs one by one and re-analyzing pooled effects. Publication bias was assessed by examining the asymmetry of funnel plots using the Egger [24] and Begg tests [25]. 

### 2.6. Certainty Assessment

The general certainty of evidence across RCTs was rated using the Grading of Recommendations Assessment, Development, and Evaluation (GRADE) working group guidelines. According to the related assessment criteria, the quality of evidence was categorized into four classes: high, moderate, low, and very low [26].

## 3. Results

### 3.1. Selection and Identification of RCTs

Out of 3282 publications initially identified via database search, 1495 duplicate articles were excluded. After screening the remaining 1787 records, an additional 1741 unrelated articles were removed based on title and abstract assessment. Then, 46 publications remained for further subsequent full-text evaluation. Three studies were excluded due to age restrictions [27,28,29]. The investigation by García-Pinillos et al. was also excluded as their participants performed their own exercise routine in addition to the main training intervention (HIIT) [30]. Moreover, two eligible articles published results on the same dataset [31,32], and the more comprehensive manuscript was included for analysis [31]. A further five studies did not indicate a control group [33,34,35,36,37], leaving a total of 36 eligible RCTs for final meta-analysis [31,38,39,40,41,42,43,44,45,46,47,48,49,50,51,52,53,54,55,56,57,58,59,60,61,62,63,64,65,66,67,68,69,70,71,72]. Of these 36 RCTs, 35 studies assessed changes in BF% [31,38,39,40,41,42,43,44,45,46,47,48,49,50,51,52,53,54,55,56,57,58,59,60,61,62,63,64,65,66,67,68,69,70,71,72] while FM and FFM changes were investigated in 14 studies [40,45,51,55,58,59,60,62,63,64,66,67,69,73] and nine studies [40,51,55,58,60,61,62,66,67], respectively. A flow diagram of the study selection process is outlined in Figure 1.

### 3.2. Characteristics of the Included RCTs

Characteristics of the 36 RCTs included in the current systematic review and meta-analysis are illustrated in Table 2. These RCTs were published in English, between 2008 and 2022 with regions of origin, including the USA, Europe, Asia, and Canada. A minority of studies included both male and female participants (n = 7), whereas 18 studies were exclusively performed with males [40,41,43,46,49,51,52,53,56,58,73] and 11 with females only [31,44,45,47,55,59,60,63,64,65,67,70]. The total sample size included 1130 individuals, 551 of whom were in a HIIT exercise intervention group, and 579 served as controls. The mean age of participants was between 18 and 57 years. According to Cochrane scores, as described, nine studies were classified as high-quality, 27 were classified as moderate-quality, and no study was deemed to be low quality. The results of the quality assessment are reported in Appendix A.

### 3.3. Findings from the Systematic Review

Among the 47 intervention arms from 36 studies, 45 arms assessed BF% changes following HIIT interventions with 30 reporting significant reductions in BF% [31,38,39,40,43,44,45,47,48,50,51,53,54,56,57,59,60,62,63,64,69,70,71,72] compared to the others which did not [41,46,49,52,54,55,57,58,61,65,66,67,68,74]. In addition, modes of HIIT exercise varied across investigations such that 17 studies utilized cycling [49,50,51,52,53,54,55,56,57,58,59,60,61,62,63,64,65,73], 10 involved overground running [31,38,39,40,41,43,44,46,47,48], and eight incorporated treadmill protocols [45,66,67,68,69,70,71,72]. Twenty-one studies assessed FM changes, of which nine reported significant FM reductions [40,45,51,55,58,59,60,62,63,64,66,67,69,73]. Regarding changes in FFM, three trials illustrated significant increases following HIIT [40,51] while six did not [55,58,60,61,62,67]. 

### 3.4. Findings from the Meta-Analysis

Overall, 47 effect sizes from 36 studies following the systematic review were included for meta-analysis. These trials had a total sample size of 1130 individuals, and the mean age was 26.27 ± 5.42 years.

**The effect of HIIT on BF%**: After combining 45 effect sizes from 35 studies (n = 1082) [31,38,39,40,41,42,43,44,45,46,47,48,49,50,51,52,53,54,55,56,57,58,59,60,61,62,63,64,65,66,67,68,69,70,71,72,73], pooled effects data analysis indicated that HIIT, compared to the control, induced a significant reduction in BF% [weighted mean difference (WMD): −1.53%, 95% CI: −2.13, −0.92, *p =* 0.001; Table 3] despite high between-study heterogeneity (I^2^ = 86.7, *p =* 0.001). Subgroup analyses revealed that while all forms of HIIT (cycling vs. overground running vs. treadmill running) reduced BF%, overground running induced the largest overall effect (WMD: −2.80%, 95% CI: −3.89 to −1.71, *p =* 0.001; Figure 2). Further, in studies performed over 8 weeks duration, a larger effect on BF% was noted when compared to shorter (i.e., less than 8 weeks) interventions (WMD: −2.08%, 95% CI: −2.40 to −1.76, *p =* 0.001). In the training frequency subgroup, it was shown that three sessions per week had a significant effect on reducing BF% (WMD: −1.17%, 95% CI: −1.85 to −0.50, *p =* 0.001), while two sessions per week were not significant (WMD: −3.23%, 95% CI: −6.52 to 0.04, *p =* 0.054). All subgroups of time of training (≤60 s and >60 s) produced a significant reduction in BF% (WMD: −1.87%, 95% CI: −2.78 to −0.96, *p =* 0.001 and WMD: −0.92%, 95% CI: −1.71 to −0.13, *p =* 0.022, respectively). The rest time subgroup (≤90 s and >90 s) showed a significant reduction in BF% (WMD: −1.94%, 95% CI: −2.83 to −1.06, *p =* 0.001 and WMD: −1.01%, 95% CI: −1.84 to −0.10, *p =* 0.017, respectively). BF% was reduced in all groups regardless of the type [active (WMD: −1.60%, 95% CI: −2.43 to −0.77, *p =* 0.001) or passive (WMD: −1.56%, 95% CI: −2.33 to −0.79, *p =* 0.001)] of rest period. Furthermore, for BMI and gender, there was a significant reduction in BF% in all subgroups, except for BMI > 30 (WMD: 0.29%, 95% CI: −0.15 to 0.73, *p* = 0.197). All results from subgroup analyses for BF% are shown in Table 3.

Sensitivity analysis revealed that excluding individual RCTs did not affect the overall estimate of HIIT on BF% (range of summary estimates: −1.41, −1.00). In addition, based on visual inspection of the funnel plot, asymmetry was apparent; however, Begg and Egger regression tests (*p =* 0.187 and *p =* 0.219, respectively) indicated no significant observable publication bias.

**The effect of HIIT on FM:** Based on the results of 14 studies [40,45,51,55,58,59,60,62,63,64,66,67,69,73] containing 21 effect sizes (n = 565), HIIT resulted in a significant reduction in FM (WMD: −1.86 kg, 95% CI: −2.55 to −1.18, *p =* 0.001, Table 3). However, there was evidence of a medium between-study heterogeneity (I^2^ = 63.3, *p =* 0.001). Subgroup analyses revealed that cycling and overground running reduced FM (WMD: −1.72 kg, 95% CI: −2.41 to −1.30, *p =* 0.001 and WMD: −4.25 kg, 95% CI: −5.90 to −2.61, *p =* 0.001, respectively, Figure 3), while treadmill running did not induce significant changes (WMD: −1.10 kg, 95% CI: −2.82 to 0.62, *p =* 0.210, Figure 3). Further, in RCTs over 8 weeks, a larger effect on FM was noted when compared to shorter (i.e., less than 8 weeks) interventions (WMD: −1.92 kg, 95% CI: −2.35 to −1.50, *p =* 0.001). The training frequency subgroup showed that 2 and 3 sessions per week had a significant effect on reducing FM (WMD: −4.43 kg, 95% CI: −6.62 to −2.24, *p =* 0.001 and WMD: −1.24 kg, 95% CI: −2.00 to −0.48, *p =* 0.001, respectively). All subgroups of time training (≤60 s and >60 s) produced a significant reduction in FM (WMD: −2.20 kg, 95% CI: −3.01 to −1.39, *p =* 0.001 and WMD: −1.14 kg, 95% CI: −2.13 to −0.15, *p =* 0.023, respectively). The rest time subgroup (≤90 s and >90 s) showed a significant reduction in FM (WMD: −2.26 kg, 95% CI: −3.08 to −1.43, *p =* 0.001 and WMD: −1.07 kg, 95% CI: −1.99 to −0.15, *p =* 0.021, respectively). FM was reduced in all groups regardless of the type [active (WMD: −2.13 kg, 95% CI: −2.99 to −1.27, *p =* 0.001) or passive (WMD: −1.52 kg, 95% CI: −2.77 to −0.27, *p =* 0.017)] of rest period. Furthermore, for BMI and gender, there was a significant reduction in FM in all subgroups, except for BMI > 30 (WMD: −0.99 kg, 95% CI: −4.58 to 2.60, *p* = 0.589). All results from subgroup analyses for FM are shown in Table 3. 

Sensitivity analysis revealed that excluding individual RCTs did not affect the overall estimate of HIIT on FM (range of summary estimates: −1.88, −1.23). In addition, based on visual inspection of the funnel plot, asymmetry was apparent; however, Begg and Egger regression tests (*p =* 0.478 and *p =* 0.193, respectively) indicated no significant observable publication bias.

**The effect of HIIT on FFM:** In total, 12 effect sizes from nine studies [40,51,55,58,60,61,62,66,67], including 327 participants, were included for meta-analysis. Upon combining the effect sizes, HIIT induced an overall significant improvement in FFM (WMD: 0.51 kg, 95% CI: 0.06 to 0.95, *p =* 0.025, Table 3) which was further emphasized by no heterogeneity among studies (I^2^ = 0.0%, *p* = 0.614).

As illustrated in Figure 4, only HIIT involving cycling interventions resulted in a significant increase in FFM compared to other exercise modalities (WMD: 0.63 kg, 95% CI: 0.17 to 1.09, *p =* 0.007). As with BF%, studies utilizing interventions longer than 8 weeks produced larger effects than trials of shorter (i.e., less than 8 weeks) durations (WMD: 0.63 kg, 95% CI: 0.17 to 1.10, *p =* 0.008). The training frequency subgroup showed that 3 sessions per week induced a significant increase in FFM (WMD: 0.50 kg, 95% CI: 0.06 to 0.94, *p =* 0.025). Regarding the duration of training per repetition, ≤60 s produced a significant increase in FFM (WMD: 0.57 kg, 95% CI: 0.29 to 1.44, *p =* 0.003) while >60 s did not induce significant changes (WMD: −0.85 kg, 95% CI: −2.28 to 0.57, *p =* 0.239). The rest time subgroup for ≤90 s and active rest showed a significant increase in FFM (WMD: 0.70 kg, 95% CI: 0.22 to 1.18, *p =* 0.004 and WMD: 0.68 kg, 95% CI: 0.20 to 1.17, *p =* 0.004, respectively); however, passive and >90 s rest had no significant increase on FFM (WMD: −0.26 kg, 95% CI: −1.44 to 0.91, *p =* 0.657, and WMD:−0.53 kg, 95% CI: −1.81 to 0.73, *p =* 0.410, respectively). Furthermore, for BMI and gender, there was a significant increase in FFM in all subgroups, except, women and BMI > 30 subgroups (WMD: −0.47 kg, 95% CI: −1.58 to 0.63, *p* = 0.404, and WMD: 0.62 kg, 95% CI: −1.90 to 3.14, *p* = 0.629, respectively). Results for all subgroup analyses on FFM are shown in Table 3. 

Sensitivity analysis revealed that after excluding individual RCTs, the removal of one study [51] in particular eliminated any observed significant overall effect of HIIT on FFM (WMD: −0.46, 95% CI: −1.38 to 0.44). Moreover, the Begg and Egger test rejected our hypothesis about the presence of substantial publication bias (*p =* 0.244 and *p =* 0.079, respectively).

### 3.5. Quality of Evidence

The GRADE guideline was employed to assess the quality of evidence for outcomes, which indicated the effect of FFM to be of high quality. However, the evidence about FM and BF% was downgraded to medium and low levels, respectively (Table 4).

## 4. Discussion

The current meta-analysis investigated whether, as a whole or defined aspect of a HIIT program (e.g., exercise modality, duration of intervention, participant sex, and BMI) had differing effects on body composition, including BF%, FM, and FFM. We observed an overall significant pooled effect that HIIT reduces BF% by 1.53% (*p* = 0.001). Subgroup analysis based on the exercise mode of HIIT showed that overground running induced the highest reduction in BF% (WMD = −2.80%; *p* = 0.001) and FM (WMD: −4.25 kg; *p* = 0.001) compared to other modalities. Additionally, FFM increased only in studies utilizing a HIIT cycling intervention (WMD = 0.63 kg; *p* = 0.007), while treadmill running and overground running did not increase FFM. Subgroup analysis investigating the intervention length revealed significant changes occurred for BF%, FM, and FFM in studies longer than 8 weeks duration, high frequency (3 sessions/week), with a work training bout duration of ≤60 s and ≤90 s, and active recovery time.

To the best of our knowledge, this is the first meta-analysis to demonstrate the positive effects of different HIIT types on measures of body composition, including BF%, FM, and FFM in an adult population. A previous meta-analysis of RCTs in overweight and obese participants (as opposed to the general population in our current work) showed significant reductions in BF% following HIIT, particularly when incorporating running modalities compared to cycling [19]. However, a significant limitation of this work included a small pool of RCTs (n = 13) and thus an inability to perform subgroup analyses, and a lack of investigation into the effects of HIIT on FFM. We observed an overall significant reduction of BF% in all modes of HIIT, and a subgroup analysis of the exercise mode (overground running, cycling, or treadmill running) showed a magnitude effect with the overground running protocol. Accordingly, we speculate that the magnitude of changes in BF% following HIIT is dependent on exercise modality, equipment utilized, and/or relative exercise intensity in healthy individuals. Indeed, our findings suggest cycling or treadmill running does not provide the same exercise stimulus (intensity) as overground running. It is plausible overground running, particularly with terrain variations (i.e., diverse surfaces and inclines/declines), may activate larger and more numerous muscle groups compared to treadmill running or stationary cycle ergometers to more effectively increase metabolic rate. Thus, while our findings suggest overground running is a preferential HIIT modality for reducing BF% in healthy adults, it should be recognized that this may not be applicable to clinical cohorts. For instance, it has been established that overweight and individuals with obesity exhibit altered joint loads, absolute ground reaction forces, and forefoot pressures compared to non-obese adults [75], indicating the biomechanical stress of overground running may be more detrimental to joint health in these individuals compared to weight-bearing activities such as cycling. Nevertheless, previous research has shown that a 1% reduction in BF% can lead to a 3% decrease in the risk of type 2 diabetes [76], and our findings suggest that HIIT may be an effective intervention for achieving these reductions. Thus, incorporating HIIT as a routine may be a promising approach with important clinical implications.

We also observed a greater magnitude decrease in FM in both the overground running and cycle groups, although the treadmill subgroup did not show a significant change, likely due to the small number of studies. In contrast, a systematic review and meta-analysis by Wewege et al. [19] found no significant effect of cycling on measures of body composition (whole-body FM) in overweight and obese adults. While the physiological basis of such findings is unclear, potential mechanisms that may explain the discrepancy between cycling and other exercise modalities include muscle recruitment patterns (e.g., cycling, as a none weight-bearing activity, may preferentially recruit slow-twitch muscle fibers), body position during exercise such as standing vs. sitting, and total caloric expenditure during HIIT. In this regard, previous work has indicated that the more muscle mass recruited during exercise at any given exercise intensity, the greater the energy expenditure [19,20], which is all the more relevant when HIIT involves repeated exercise bouts of near or at maximal intensities. Moreover, the skeletal muscle pump, which facilitates venous return to the heart and intravenous injection of skeletal muscle, has been shown to be more efficient during running than cycling [20]. Several factors influence the activity of the skeletal muscle pump (and thus blood flow), including the frequency of contraction, which is directly influenced by the rate of running steps or cycling revolutions per minute [20]; body position, where standing promotes involuntary muscle contraction vs. sitting; and type of muscle contraction, such as the stretch-shortening cycle during exercise [77]. All these factors are seemingly advantageous in weight-bearing activities like running. Further, the greater muscle mass involvement observed during running leads to increased blood flow which may play a role in increasing the rate of fat oxidation and fat oxidation kinetics, ultimately leading to more significant reductions in FM and BF% over time [78]. In addition, the kinematics of overground and treadmill running are different such that treadmill platforms provide a lower impact stimulus and potentially less muscle activation compared to overground running. For example, the *rectus femoris* and *biceps femoris* muscles of the anterior and posterior regions of the quadriceps, respectively, have been shown by electromyography (EMG) to be more active during overground running than in treadmill running [79]. While such enhanced lipid metabolism may be due to the exercise stimulus itself, it is likely that even though total energy expenditure is relatively low during HIIT training, greater reductions in body fat come from elevated post-exercise oxygen consumption (EPOC) observed with activities involving more muscle mass [80]. Future investigations should endeavor to determine the most effective HIIT exercise modality under well-controlled and similar conditions (e.g., intervention duration, relative/absolute intensity, participant characteristics, etc.), notwithstanding any participant limitations.

Overall, HIIT appeared to significantly and effectively influence FFM, yet such differences were only noted in RCTs utilizing a cycling intervention (WMD = 0.63 kg; *p* = 0.007) compared to other exercise modalities investigated. The reasons for such disparity in the response of FFM to exercise modality are not fully known. Intuitively, this may be due to the relative specificity of recruiting large muscle groups of the lower extremity necessary for pedaling [20,81] and resistance imposed by increases in cycle ergometer wattage/power output that are distinct from the gravitational forces imposed by running. Heydari et al. [51] similarly observed significant increases in FFM following 12 weeks of HIIT in young males, while we have also demonstrated 6 weeks of cycle ergometer HIIT to significantly increase lean mass (1.0 kg), albeit in conjunction with a high protein diet [82]. More research is required on the possible mechanism of enhanced cycling-induced FFM improvements during HIIT compared to other modalities before definitive conclusions may be reached.

Moreover, subgroup analysis in our current study indicated that a more than 8 weeks duration of HIIT proves more beneficial for BF%, FM, and FFM than shorter duration programs, indicating the need for a prolonged stimulus to reduce body fat and promote muscle mass remodeling. It was also found that three sessions per week were more effective on body composition than two sessions per week in the training frequency subgroup. Consistent with the findings of a study by Stavrinou et al. [57], it was concluded that a higher frequency of exercise during a week has effective results in reducing BF%. There is evidence that high-frequency exercise (≥3 sessions) can promote fat loss due to significant catecholamine response, increased concentration of β-adrenergic receptors, and increased fat oxidation versus low frequency [57]. Our analysis of the existing data revealed that a lower time of training (≤60 s) per repetition and shorter rest (≤90 s) period was more effective for improving body composition than a longer training schedule. A short work bout duration of ≤60 s per repetition increased FFM while a longer work bout (>60 s) had no significant effect. In agreement with our finding, multiple studies with short-term intervals have reported reduced body fat [14,63,83,84]. However, a recent study showed both short and long intervals reduced BF% and FM (kg), although short-term intervals (≤60 s) induced a greater reduction. In this regard, Rønnestad et al. [85] reported that the short-intervals protocol (30 s work intervals separated by 15 s recovery) achieved a larger relative improvement in VO_2max_ than the long intervals group (4 × 5-min work intervals separated by 2.5-min recovery periods). Another novel finding from our current work was that active and short rest (≤90 s) is more effective for promoting body composition changes, especially FFM. Both active and passive recovery showed a reduction in BF% and FM. There is a paucity of knowledge regarding the mechanistic basis for these positive effects on body composition with active and passive recovery. Findings by Spencer et al. [86] showed active recovery promotes an increase in muscle lactate removal compared to the passive recovery condition. Presumably, in active recovery, blood lactate concentration was decreased due to an increase in muscle lactate metabolism, rather than a greater release of lactate. High blood lactate levels down-regulate the use of glucose and free fatty acids (FFA). Furthermore, lactate accumulation upregulates the expression of monocarboxylate transporter 1 (MCT1) which may serve a role in the efflux of lactate [87]. Nonetheless, in our results, regardless of rest mode, BF% and FM (kg) decreased significantly, while only active rest could increase FFM (WMD: 0.68; *p* = 0.004). Future studies are required to better understand the effect of HIIT with active recovery on FFM changes.

Several limitations in the present meta-analysis are worth noting when interpreting our results. Primarily, a majority of investigations were conducted on overweight or obese inactive adults as opposed to active normal body mass or BMI adults. Hence, it remains challenging to draw generalized conclusions on how different types of HIIT influence body composition outcomes in these populations (active and inactive adults). In addition, significant heterogeneity (I^2^) was found across studies, particularly with BF%, as noted, which likely relates to the large variability in study design (e.g., exercise modalities, duration, participant sex, age range, and training intensity) between studies. Furthermore, BF%, FM, and FFM were measured using different methodologies (e.g., subcutaneous skinfold caliper, ADP, BIA, and DEXA), pre-test guidelines, dietary control, participant hydration status, etc., which may give different results when tracking the changes in body composition variables [88]. 

In conclusion, the results of this meta-analysis demonstrate favorable body composition outcomes following HIIT, including overall reductions in BF%, FM, and improved FFM. Such improvements in BF% were enhanced following HIIT involving overground running compared to other exercise modalities. While similar overall improvements as BF% were noted for FFM following HIIT, in this case, cycling proved to be the only effective mode of HIIT. These findings indicate that individuals who want to reduce their BF% may benefit more from overground running, while those who want to increase their FFM may benefit more from cycling. In all cases, studies lasting 8 weeks or more provided the necessary stimulus to promote improvements in BF%, FM, and FFM when pooled together. These data suggest that training for more than 8 weeks, at least three sessions per week, lower than 60 s work intervals separated by ≤90 s recovery periods, and active rest (instead of passive) are more effective on body composition. In light of the public health implications of maintaining proper body composition for health and well-being, and considering the increasing popularity of HIIT exercises, future research should aim to determine ideal training models for maximizing improvements in body fat and FFM in the general population.

## Figures and Tables

**Figure 1 jcm-12-02291-f001:**
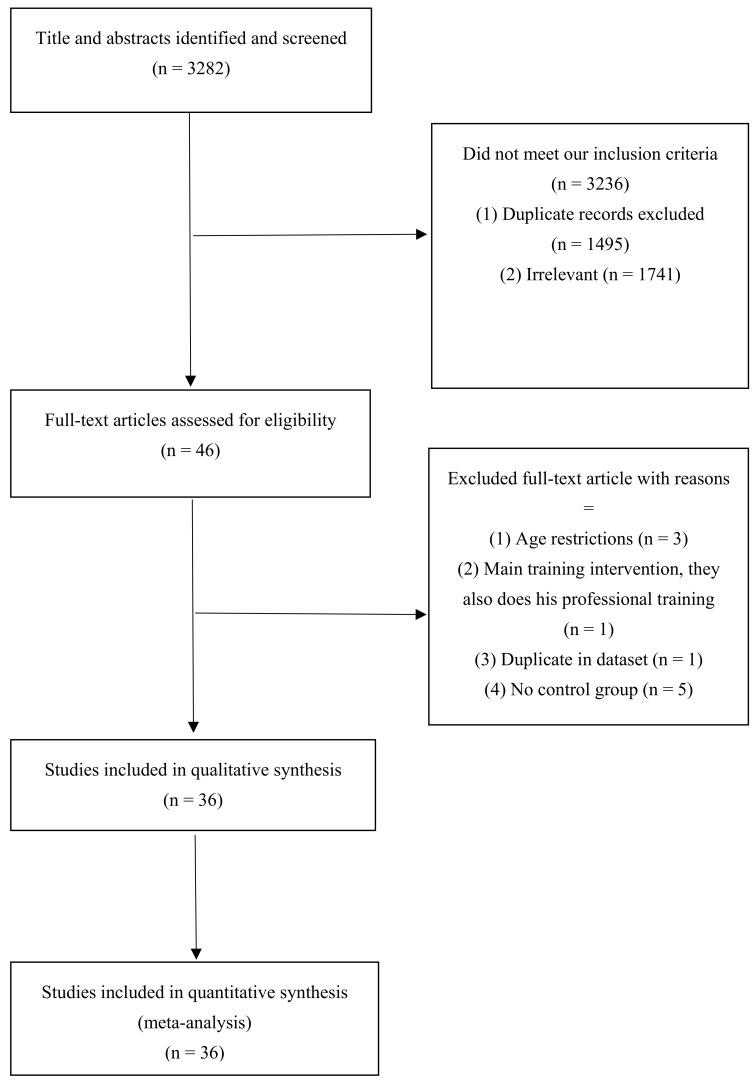
Flowchart of study selection for inclusion trials in the systematic review.

**Figure 2 jcm-12-02291-f002:**
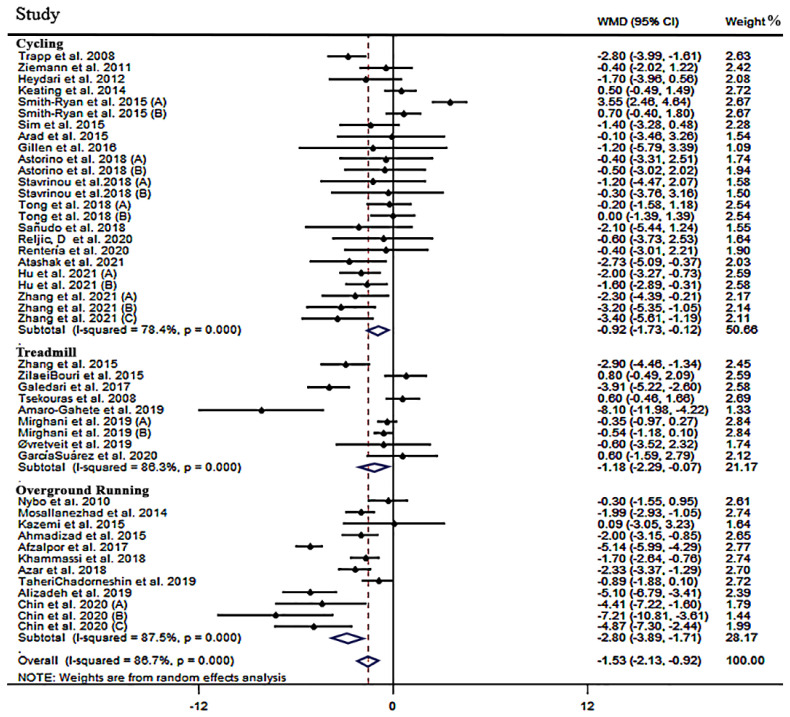
Forest plot detailing weighted mean difference and 95% confidence intervals (CIs) for the effect of high-intensity interval training on BF%.

**Figure 3 jcm-12-02291-f003:**
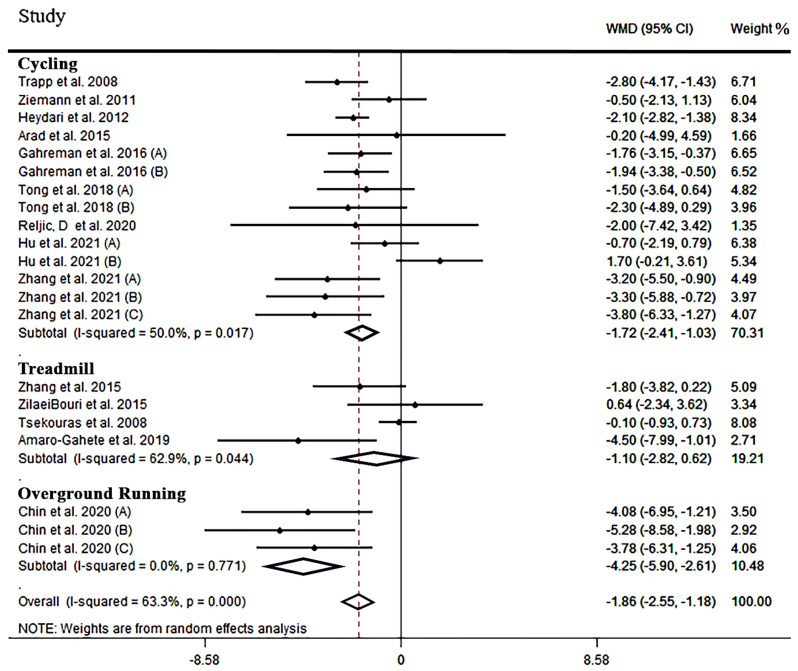
Forest plot detailing weighted mean difference and 95% confidence intervals (CIs) for the effect of high-intensity interval training on FM.

**Figure 4 jcm-12-02291-f004:**
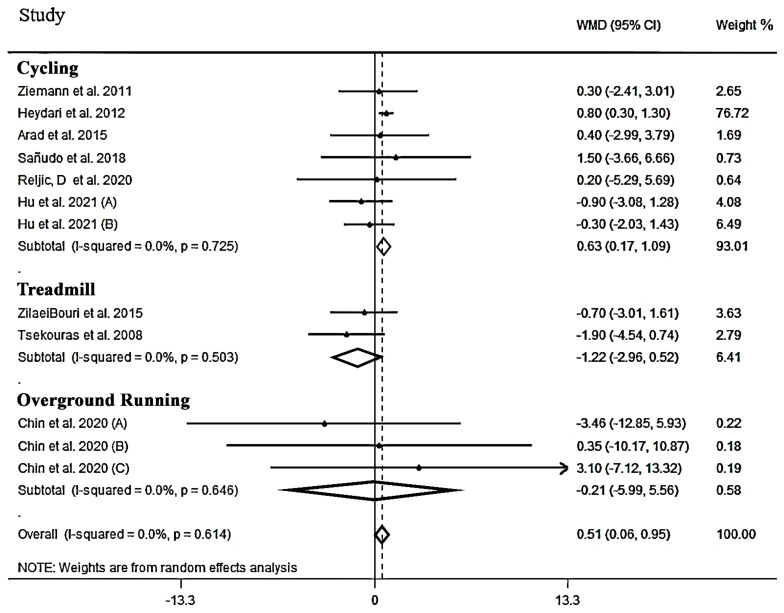
Forest plot detailing weighted mean difference and 95% confidence intervals (CIs) for the effect of high-intensity interval training on FFM.

**Table 1 jcm-12-02291-t001:** Participants, intervention, comparators, outcomes, study design (PICOS) criteria for inclusion of studies.

Population	Adult Population Cohorts (≥18 Years)
Intervention	Exercise training (HIIT) involving either cycling, overground running, or treadmill running
Comparison	Between HIIT modes and non-exercise control group
Outcomes	Body fat percentage, fat mass, and fat-free mass
Study design	Human randomized control trials

**Table 2 jcm-12-02291-t002:** Characteristics of the RCTs included in the current systematic review and meta-analysis.

Study	Participants	No. (Control/Intervention)	Mean Age	BMI	Duration	Type of HIIT	Exercise Intervention	BF%	FFM (kg)	FM (kg)	Equipment
Sim et al., 2015 [52]	Overweight inactive men	20 10/10	27.2 ± 1.2	31 ± 8	3 d/12 wk	Cycling	15 s at a power output equivalent to approximately 170% VO_2peak_ with an active recovery period (60 s at a power output of approximately 32% VO_2peak_) between efforts	↔	NR	NR	DEXA
Keating et al., 2014 [50]	Inactive, overweight adult	26 13/13	42.9 ± 2.3	28.3 ± 0.5	3 d/12 wk	Cycling	1–12 sets of 30–60 s at 120% VO_2peak_ with 120–180 s active recovery period	↓	NR	NR	DEXA
Smith-Ryan et al., 2015 (A) [49]	Overweight men	25 10/5	38.3 ± 11.5	31.3 ± 4.9	3 d/3 wk	Cycling	10 sets × 1 min at 90% VO_2peak_ with 60 s active recovery period	↔	NR	NR	DEXA
Smith-Ryan et al., 2015 (B) [49]	Overweight men	25 10/5	38.3 ± 11.5	31.3 ± 4.9	3 d/3 wk	Cycling	5 sets of 2 min at 80–100% VO_2peak_ with 60 s active recovery period	↔	NR	NR	DEXA
Heydari et al., 2012 [51]	Inactive, overweight men	46 21/25	24.9 ± 4.3	28.7 ± 0.7	3 d/12 wk	Cycling	8 s sprint, 12 s recovery, continuously throughout each 20-min session.	↓	↑	↓	DEXA
Gillen et al., 2016 [53]	Sedentary men	15 6/9	27 ± 8	26 ± 6	3 d/12 wk	Cycling	3 sets × 20 s ‘all-out’ cycle sprints (~500 W) followed by 2 min of Cycling recovery	↓	NR	NR	ADP
Astorino et al., 2018 (A) [54]	Active men and women	45 32/13	24.1 ± 5.8	NR	20 sessions/7 wk	Cycling	10 sessions of low-volume HIIT and 10 sessions of sprint 8–12 sets 30–60 s at 90–150% PPO sprints Followed by 75–120 recovery	↔	NR	NR	Caliper
Astorino et al., 2018 (B) [54]	Active men and women	45 32/13	23.8 ± 3.8	NR	20 sessions/7 wk	Cycling	10 sessions of low-volume HIIT and 10 sessions of HIIT, 5–10 sets 60–150 s at 70–110% PPO sprints Followed by 60–75 recovery	↓	NR	NR	Caliper
Arad et al., 2015 [55]	Healthy, pre-menopausal	28 11/9	29.5 ± 5.5	32.3 ± 3.4	3 d/14 wk	Cycling	Four work intervals (30–60 s at 75–90% HRR) were performed with recovery intervals (180–210 s at 50% HRR) interspersed.	↔	↔	↔	DEXA
Atashak et al., 2021 [56]	Healthy but inactive males with obesity	30 15/15	24.9 ± 3.1	30.9 ± 1.04	3 d/12 wk	Cycling	5 × 2 min interval bout at an intensity of 85–95% HR max interspersed by 1 min passive recovery, three times per week	↓	NR	NR	Caliper
Stavrinou et al., 2018 (A) [57]	Healthy inactive adults	22 8/13	31.8 ± 1.6	23.7 ± 3.6	3 d/8 wk	Cycling	10 × 60 s cycling intervals at an intensity of ~83% of the Wpeak obtained, interspersed With 60 s of low-intensity exercise (~30% Wpeak at 50 rpm).	↓	NR	NR	Caliper
Stavrinou et al., 2018 (B) [57]	Healthy inactive adults	22 8/14	31.6 ± 2.15	23.5 ± 3.8	2 d/8 wk	Cycling	10 × 60 s cycling intervals at an intensity of ~83% of the Wpeak obtained, interspersed With 60 s of low-intensity exercise (~30% Wpeak at 50 rpm).	↔	NR	NR	Caliper
Ziemann et al., 2011 [58]	Healthy, physically active but not highly trained, college-aged men	21 11/10	21.3 ± 1	23.7 ± 1.85	3 d/6 wk	Cycling	6 × 90 s bouts at 80% of VO_2max_ (each followed by 180 s passive recovery)	↔	↔	↔	BIA
Tong et al., 2018 (A) [59]	Female university students	30 14/16	21 ± 1.2	NR	3 d/12 wk	Cycling	80 × 6 s all-out cycle sprints interspersed with 9 s passive recovery	↓	NR	↔	DEXA
Tong et al., 2018 (B) [59]	Female university students	30 14/16	21 ± 1.2	NR	3 d/12 wk	Cycling	Repeated 4 min bouts of cycling at 90% VO_2max_ alternated with 3 min passive recovery until the work of 400 KJ was achieved	↓	NR	↔	DEXA
Hu et al., 2021 (A) [60]	Overweight/obese females	30 15/15	21.2 ± 1.4	25.7 ± 2.4	3 d/12 wk	Cycling	4 min cycling at 90% VO_2peak_ followed by 3 min passive recovery for ~60 min	↓	↔	↓	DEXA
Hu et al., 2021 (B) [60]	Overweight/obese females	30 15/15	21.1 ± 1.05	25.7 ± 2.3	3 d/12 wk	Cycling	80 × 6 s “all-out” Cycling interspersed with 9 s passive recovery	↓	↔	↓	DEXA
Sañudo et al., 2018 [61]	Obese/overweight adults	27 13/14	36.5 ± 8	31.7 ± 5.2	3 d/8 wk	Cycling	6–10 sets × 1 min of HIIT at 90% HRpeak followed by 6–10 × 2 min passive recovery	↔	↔	NR	BIA
Reljic, D et al., 2020 [62]	Obese individuals with a sedentary occupation	49 19/30	48.7 ± 9.9	39.4 ± 7.05	2 d/12 wk	Cycling	5 interval bouts of 1 min at 80–95% HRmax interspersed with 1 min of low-intensity recovery	↓	↔	↓	BIA
Trapp et al., 2008 [63]	Inactive but healthy women	30 15/15	22.3 ± 0.4	24.1 ± 1.4	3 d/15 wk	Cycling	60 sets of 8 s at a resistance of 0.5 kg and worked as hard as they could followed by 12 s of slow Cycling recovery	↓	NR	↓	DEXA
Zhang et al., 2021 (A) [64]	Obese young women	24 13/11	21.05 ± 1.9	25.4 ± 2.1	44 se/12 wk	Cycling	40 bouts of 6 s all-out SIT (SIT all-out) interspersed with 9 s passive recovery	↓	NR	↓	DEXA
Zhang et al., 2021 (B) [64]	Obese young women	25 13/12	20.4 ± 1.7	25.6 ± 2.5	44 se/12 wk	Cycling	Supramaximal SIT (SIT120) The total work done per training session was confined to 200 KJ 1-min exercise bouts at the work rate corresponding to 120% VO2peak, interspersed with 1.5-min passive recovery intervals	↓	NR	↓	DEXA
Zhang et al., 2021 (C) [64]	Obese young women	25 13/12	20.4 ± 1.6	25.6 ± 2.3	44 se/12 wk	Cycling	Submaximal HIIT (HIIT90) The total work done per training session was confined to 200 KJ 4-min exercise bouts at the work rate corresponding to 90% VO2peak, interspersed with 3-min passive recovery intervals	↓	NR	↓	DEXA
Rentería et al., 2020 [65]	Healthy young adult women	17 8/9	21.5 ± 1.2	24.1 ± 1.8	3 d/4 wk	Cycling	3–5 sets × 30 s HIIT at 80% maximal aerobic power, followed by 4 min of recovery at 40% MAP	↔	NR	NR	BIA
Gahreman et al., 2016 (A) [73]	Overweight males	24 12/12	26.1 + 0.7	27.9 ± 0.7	3 d/12 wk	Cycling	Green tea plus interval sprinting exercise (consumed three GT capsules daily) 60 sets of 8 s at 85% to 90% heart rate (high-intensity cycling) followed by 12 s of slow cycling recovery	NR	NR	↓	DEXA
Gahreman et al., 2016 (B) [73]	Overweight males	24 12/12	26.1 + 0.7	29.09 ± 1.04	3 d/12 wk	Cycling	Interval sprinting exercise 5 min warm-up, 20 min of ISE, 5-min cool-down 60 sets of 8 s at 85% to 90% heart rate (high-intensity cycling) followed by 12 s of slow cycling recovery	NR	NR	↓	DEXA
**Overground running**											
Ahmadizad et al., 2015 [48]	Sedentary overweight men	20 10/10	25 ± 1	27.6 ± 1.9	3 d/6 wk	Overground running	Eight exercise intervals per session with 2–3 min of active rest (rest/exercise ratio was 2:1) 90% VO_2max._	↓	NR	NR	BIA
Nybo et al., 2010 [46]	Untrained men	19 11/8	33.5 ± 2.5	NR	3 d/12 wk	Overground running	Five intervals of 2 min of near-maximal running (HR above 95% of their HRmax at the end of the 2-min period)	↔	NR	NR	DEXA
Kazemi et al., 2015 [41]	Young wrestlers	20 10/10	20–25	NR	3 d/6 wk	Overground running	3 sets of RAST protocol (6 efforts in the 35 m distance followed by a 10 s rest interval after each effort) with 4 min rest after each set in the first week. Each week one set was added to the protocol for 4 weeks.	↔	NR	NR	Caliper
TaheriChadorneshin et al., 2019 [47]	Overweight, healthy, and young women	28 14/14	30.03 ± 3.13	27.9 ± 2.9	3 d/8 wk	Overground running	4–6 sets 30 s with maximum speed and then walked for 30 s. Training progression was implemented by increasing one repetition every 2 weeks and in the 6th week, it reached 6 repetitions	↓	NR	NR	BIA
Khammassi et al., 2018 [43]	Healthy untrained overweight/obese males	20	18–21	29.1 ± 2.3	3 d/12 wk	Overground running	30 s of work at 100% MAV interspersed by 30 s of active recovery at 50% MAV, starting with 15 repetitions to reach 27 by the end of the program	↓	NR	NR	Caliper
Chin et al., 2020 (A) [40]	Overweight or obese adults	28 14/14	22.8 ± 3.1	26.4 ± 2.9	3 d/8 wk	Overground running	12 bouts × 1 min of high-intensity exercise at 90% HRR and was interspersed with 11 bouts × 1 min of active recovery at 70% HRR.	↓	↔	↓	BIA
Chin et al., 2020 (B) [40]	Overweight or obese adults	24 14/10	22.8 ± 3.1	26.4 ± 2.9	2 d/8 wk	Overground running	12 bouts × 1 min of high-intensity exercise at 90% HRR and was interspersed with 11 bouts × 1 min of active recovery at 70% HRR.	↓	↑	↓	BIA
Chin et al., 2020 (C) [40]	Overweight or obese adults	23 14/9	22.8 ± 3.1	26.4 ± 2.9	1 d/8 wk	Overground running	12 bouts × 1 min of high-intensity exercise at 90% HRR and was interspersed with 11 bouts × 1 min of active recovery at 70% HRR.	↓	↑	↓	BIA
Afzalpour et al., 2017 [31]	Overweight women	20 10/10	21.1 ± 1.4	27.5 ±1.2	3 d/10 wk	Overground running	4–8 sets 30 s at 85–95% HR max followed by 30 s active rest	↓	NR	NR	Caliper
Alizadeh et al., 2019 [38]	Overweight adolescent boys	20 10/10	18 ± 1.5	27.6 ± 0.8	3 d/6 wk	Overground running	4–6 sets 30 s at 90% of HR max followed by 30 s active rest	↓	NR	NR	Caliper
Azar et al., 2018 [39]	Sedentary young men	18 9/9	23.8 ± 1.7	23.4 ± 2.4	3 d/6 wk	Overground running	Each session consisted of either four to six repeats of maximal sprint running within a 20 m area with 20–30 s recovery.	↓	NR	NR	Caliper
Mosallanezhad et al., 2014 [44]	Inactive normal young women	21 10/11	23.8 ± 1.6	23.7 ± 4.3	3 d/8 wk	Overground running	3–6 times of running with maximum speed in a 20-m area with 30 s rest from each other	↓	NR	NR	NR
**Treadmill running**											
Tsekouras et al., 2008 [66]	Young nonobese men	15 8/7	20–40	24.3 ± 0.9	3 d/8 wk	Treadmill	Subjects alternated four times between 4 min at 60% of pre-training VO_2peak_ and 4 min at 90% of pre-training VO_2peak_ for a total of 32 min	↔	↔	↔	DEXA
Zhang et al., 2015 [45]	Overweight women	23 11/12	20.9 ± 1	25.6 ± 2.1	4 d/12 wk	Treadmill	4 × 4-min running at 85–95% HRpeak, interspersed by 3-min walking at 50–60% HRpeak	↓	NR	↓	BIA
GarcíaSuárez et al., 2020 [68]	Physically active male	19 10/9	21.5 ± 1.6	22.8 ± 2.05	3 d/over 4 wk	Treadmill	The initial three sessions started with a 2 min run warm-up at 40% VO_2peak_. Then, a high-intensity interval was performed for 2 min at 100% VO_2peak_, for a total of three high-intensity and low-intensity bouts	↔	NR	NR	BIA
ZilaeiBouri et al., 2015 [67]	Obese and overweight female	14 7/7	23.1 ± 2.6	29.1 ± 2.3	3 d/over 8 wk	Treadmill	4 × 4 min intervals at 85–95% peak heart rate, separated by 3 × 3 min of active recovery at 50–70% peak heart rate	↔	↔	↔	NR
Amaro-Gahete et al., 2019 [69]	Middle-aged adults men and women	30 14/16	52.7 ± 4.9	26.5 ± 3.5	2 d/12 wk	Treadmill	The training volume was 40–65 min/week at >95% of the maximum oxygen uptake	↓	NR	↓	DEXA
Mirghani et al., 2015 (A) [74]	Overweight to obese low active volunteer women	16 8/8	34 ± 5.3	30.1 ± 2.4	3 d/4 wk	Treadmill	4–10 set 60/60 s activity-rest at 80% reserved heart rate	↓	NR	NR	Caliper
Mirghani et al., 2015 (B) [74]	Overweight to obese low active volunteer women	16 8/8	33.5 ± 5.3	28.1 ± 2.2	3 d/4 wk	Treadmill	4–10 set 60/30 s activity-rest at 80% reserved heart rate	↓	NR	NR	Caliper
Øvretveit et al., 2019 [71]	Active males	12 6/6	30.3 ± 4.0	NR	2 d/6 wk	Treadmill	4 × 4-min intervals at 85–95% of HRmax separated by 3 min of active recovery at 70% of HRmax on a Treadmill	↓	NR	NR	BIA
Galedari et al., 2017 [72]	40 non-trained overweight men	18 8/10	31.7 ± 7.2	29.4 ± 1.9	3 d/12 wk	Treadmill	6–12 × 1 min intervals running on a Treadmill at 90–95% maximal heart rate with 1 min of active rest between the intervals	↓	NR	NR	DEXA

Abbreviations: ADP: air displacement plethysmography, BIA: bioelectrical impedance analysis, d: day, DEXA: A Dual Energy X-ray Absorptiometry, HR max: maximum heart rate, HRR: heart rate reserve, min: minutes, MAP: maximal aerobic power, NR: non-report, PPO: peak power output, RAST: Running-Based Anaerobic Sprint Test, s: seconds, wk: week, Wpeak: was determined as the power of the last completed stage.

**Table 3 jcm-12-02291-t003:** Subgroup analysis of HIIT on BF%, FM, and FFM.

BF%						
Subcategories	Effect Size, n	I^2^ (%)	P-Heterogeneity	WMD	(95% CI)	*p*-Value
**Type**						
Cycling	24	78.4	0.001	−0.92	−1.73 to −0.12	0.025
Overground Running	12	87.5	0.001	−2.80	−3.89 to −1.71	0.001
Treadmill Running	9	86.3	0.001	−1.18	−2.29 to −0.07	0.037
Pooled	45	86.7	0.001	−1.53	−2.13 to −0.92	0.001
**Duration**						
≤8 wk	24	85.4	0.001	−0.62	−0.89 to −0.36	0.001
>8 wk	21	84.1	0.001	−2.08	−2.40 to −1.76	0.001
**Frequency**						
2 sessions/wk	5	79.0	0.001	−3.23	−6.52 to 0.04	0.054
3 sessions/wk	33	88.7	0.001	−1.17	−1.85 to −0.50	0.001
**Time of training** (per repetition)						
≤60 s	27	89.4	0.001	−1.87	−2.78 to −0.96	0.001
>60 s	13	70.0	0.001	−0.92	−1.71 to −0.13	0.022
**Rest time**						
≤90 s	26	89.9	0.001	−1.94	−2.83 to −1.06	0.001
>90 s	13	66.3	0.001	−1.01	−1.84 to −0.10	0.017
Active	30	89.7	0.001	−1.60	−2.43 to −0.77	0.001
Passive	10	47.6	0.046	−1.56	−2.33 to −0.79	0.001
**Gender**						
Women	17	87.8	0.001	−1.44	−1.72 to −1.17	0.001
Men	20	88.9	0.001	−1.03	−1.36 to −0.69	0.001
**BMI (kg·m^−2^)**						
<25	9	72.5	0.001	−1.33	−1.79 to −0.86	0.001
25–30	21	88.4	0.001	−1.96	−2.25 to −1.67	0.001
>30	8	86.1	0.001	0.29	−0.15 to 0.73	0.197
**FM (kg)**						
**Subcategories**	**Effect Size, n**	**I^2^ (%)**	**P-Heterogeneity**	**WMD**	**(95% CI)**	***p*-Value**
**Type**						
Cycling	14	50.0	0.017	−1.72	−2.41 to −1.30	0.001
Overground Running	4	0.0	0.771	−4.25	−5.90 to −2.61	0.001
Treadmill Running	3	62.9	0.044	−1.10	−2.82 to 0.62	0.210
Pooled	21	63.3	0.001	−1.86	−2.55 to −1.18	0.001
**Durations**						
≤8 wk	6	76.4	0.001	−0.80	−1.46 to −0.14	0.018
>8 wk	15	45.1	0.030	−1.92	−2.35 to −1.50	0.001
**Frequency**						
2 sessions/wk	3	0.0	0.598	−4.43	−6.62 to −2.24	0.001
3 sessions/wk	13	66.3	0.001	−1.24	−2.00 to −0.48	0.001
**Time of training** (per repetition)						
≤60 s	13	54.2	0.010	−2.20	−3.01 to −1.39	0.001
>60 s	6	52.1	0.064	−1.14	−2.13 to −0.15	0.023
**Rest time**						
≤90 s	12	57.0	0.007	−2.26	−3.08 to −1.43	0.001
>90 s	7	42.8	0.106	−1.07	−1.99 to −0.15	0.021
Active	11	64.4	0.002	−2.13	−2.99 to −1.27	0.001
Passive	8	65.5	0.005	−1.52	−2.77 to −0.27	0.017
**Gender**						
Women	11	60.1	0.005	−1.62	−2.25 to −0.99	0.001
Men	8	73.8	0.001	−1.53	−1.97 to −1.08	0.001
**BMI (kg·m^−2^)**						
<25	3	81.9	0.004	−0.78	−1.43 to −0.13	0.019
25–30	14	62.0	0.001	−1.97	−2.42 to −1.52	0.001
>30	2	0.0	0.626	−0.99	−4.58 to 2.60	0.589
**FFM (kg)**						
**Subcategories**	**Effect Size, n**	**I^2^ (%)**	**P−Heterogeneity**	**WMD**	**(95% CI)**	***p*-Value**
**Type**						
Cycling	7	0.0	0.725	0.63	0.17 to 1.09	0.007
Overground Running	3	0.0	0.646	−0.21	−5.99 to 5.56	0.942
Treadmill Running	2	0.0	0.503	−1.22	−2.96 to 0.52	0.169
Pooled	12	0.0	0.614	0.50	0.06 to 0.94	0.025
**Durations**						
≤8 wk	7	0.0	0.822	−0.58	−1.95 to 0.78	0.402
>8 wk	5	0.0	0.481	0.63	0.17 to 1.10	0.008
**Frequency**						
2 sessions/wk	2	0.0	0.980	0.23	−4.63 to 5.10	0.926
3 sessions/wk	9	9.4	0.357	0.50	0.06 to 0.94	0.026
**Time of training** (per repetition)						
≤60 s	8	0.0	0.923	0.57	0.29 to 1.44	0.003
>60 s	6	0.0	0.521	−0.85	−2.28 to 0.57	0.239
**Rest time**						
≤90 s	6	0.0	0.786	0.70	0.22 to 1.18	0.004
>90 s	5	0.0	0.664	−0.53	−1.81 to 0.73	0.410
Active	7	0.0	0.556	0.68	0.20 to 1.17	0.004
Passive	4	0.0	0.815	−0.26	−1.44 to 0.91	0.657
**Gender**						
Women	4	0.0	0.924	−0.47	−1.58 to 0.63	0.404
Men	6	0.0	0.425	0.69	0.20 to 1.17	0.006
**BMI (kg·m^−2^)**						
<25	2	23.2	0.254	−0.83	−2.72 to 1.06	0.391
25–30	7	0.0	0.467	− 0.58	0.12 to 1.04	0.014
>30	3	0.0	0.928	0.62	−1.90 to 3.14	0.629

Abbreviations: BF%, body fat percentage; FM, fat mass; FFM, fat-free mass; WMD, weighted mean difference; BMI, body mass index; s, second; wk, week.

**Table 4 jcm-12-02291-t004:** GRADE profile of the effect of high-intensity interval training type on body fat percentage and fat-free mass.

Quality Assessment						Summary of Findings			Quality of Evidence
Outcomes	Risk of bias	Inconsistency	Indirectness	Imprecision	Publication Bias	Number of Intervention/Control	WMD (95% CI)	Heterogeneity (I^2^)	
BF%	No serious limitations	Very serious Limitations	No serious limitations	No serious limitations	No serious limitations	527/555	−1.53 (−2.13, −0.92)	86.7%	⊕⊕◯◯ Low
FM	No serious limitations	Serious Limitations	No serious limitations	No serious limitations	No serious limitations	280/285	−1.86 (−2.55,−1.18)	63.3%	⊕⊕⊕◯ Medium
FFM	No serious limitations	No serious limitations	No serious limitations	No serious limitations	No serious limitations	165/162	0.51 (0.06, 0.95)	0.0%	⊕⊕⊕⊕ High

Abbreviations. BF%, body fat percentage; FM, fat mass; FFM, fat-free mass.

## Data Availability

Data sharing is applicable.

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
