# Peer review of "The Effect of High-Intensity Interval Training Type on Body Fat Percentage, Fat and Fat-Free Mass: A Systematic Review and Meta-Analysis of Randomized Clinical Trials"

_jcm, 2023, doi:10.3390/jcm12062291_

Round 1
Reviewer 1 Report (New Reviewer)
I commend the authors for submitting an interesting and well-written manuscript and thank you for offering me the opportunity to review the manuscript.
The manuscript is well-written and clear. However, there are a few small editing issues throughout the paper. For example, some sentences should be separated by a comma or semicolon. I am sure that the authors and the editorial board can review the manuscript again.
I have listed a few small comments below to improve the manuscript which should be considered.
- Line 47: I suggest using HIIT as opposed to HIT. Note that there is no HIT is used throughout the paper.
- Line 112: There is excessive space between ‘overal effect size,’ and (4).
- Line 119: I don’t think calling individuals under 18 ‘children’ is correct. Please revise and use a more appropriate word.
- Line 131-134: It is fine to contact the authors for more information. But, could you please include the number of instances that you had to do so?
- Line 97: Were all the published articles in ‘English’? Surely, there are papers published in other languages. If you have only included papers written in ‘English’ please clarify that in your methods section.
- Results: If running on the treadmill did not change the FM nor the FFM, how has it resulted in a significant reduction of BF%? This should also be discussed in the discussion section of the manuscript.
- Line 270: after analysis of the subgroups, the results have been reported in a somewhat confusing way. For example, in line 270 it is stated that ‘Active and passive rest between repetitions reduced FM’. I understand your message but it has not been written properly. It could be reported as ‘FM was reduced in all groups regardless of the type of rest period (active or passive’. Alternatively, it could be: ‘the type of rest (active or passive) did not impact the reduction of FM and participants with either type of rest experienced a reduction in FM’. Note that this approach has occurred in many parts of this manuscript. Therefore, if you decided to take action, please review the entire paper and rephrase where needed.
- Line 273: Change ‘Except’ to ‘except’
- Line 414-416: This sentence needs to be modified as your manuscript is a review not an original study. You may say ‘our analysis of the existing data revealed that a lower time of training …..
Author Response
I commend the authors for submitting an interesting and well-written manuscript and thank you for offering me the opportunity to review the manuscript.
The manuscript is well-written and clear. However, there are a few small editing issues throughout the paper. For example, some sentences should be separated by a comma or semicolon. I am sure that the authors and the editorial board can review the manuscript again.
Authors: Thank you for the feedback and your time spent reviewing our work. We are pleased to hear you feel this work may have useful applications, and we appreciate your recommendations for further improvement.
I have listed a few small comments below to improve the manuscript which should be considered.
- Line 47: I suggest using HIIT as opposed to HIT. Note that there is no HIT is used throughout the paper.
Authors: Good point. Corrections were made.
- Line 112: There is excessive space between ‘overal effect size,’ and (4).
Authors: It has been corrected.
- Line 119: I don’t think calling individuals under 18 ‘children’ is correct. Please revise and use a more appropriate word.
Authors: Good point. Other than children, we have now included the term teenagers (13-17 years).
- Line 131-134: It is fine to contact the authors for more information. But, could you please include the number of instances that you had to do so?
Authors: Thank you for raising this point; in a couple of instances. This is now included.
- Line 97: Were all the published articles in ‘English’? Surely, there are papers published in other languages. If you have only included papers written in ‘English’ please clarify that in your methods section.
Authors: All included papers were published in English, but we had no language restrictions (this was mentioned in the methods section). We have now included the English language information in the results section “Characteristics of the included RCTs”.
- Results: If running on the treadmill did not change the FM nor the FFM, how has it resulted in a significant reduction of BF%? This should also be discussed in the discussion section of the manuscript.
Authors: We would like to kindly mention that the 3 measurements were not evaluated in all studies. Specifically, 9 studies evaluated BF% after treadmill running (Figure 2A) while 4 studies (Figure 2B) evaluated FM and 2 studies (Figure 2C) FFM after treadmill running. Consequently, the lack of significance in FM and FFM after the treadmill in our results may be a result of the lack of available studies evaluating these components. This was mentioned in line 268.
- Line 270: after analysis of the subgroups, the results have been reported in a somewhat confusing way. For example, in line 270 it is stated that ‘Active and passive rest between repetitions reduced FM’. I understand your message but it has not been written properly. It could be reported as ‘FM was reduced in all groups regardless of the type of rest period (active or passive)’. Alternatively, it could be: ‘the type of rest (active or passive) did not impact the reduction of FM and participants with either type of rest experienced a reduction in FM’. Note that this approach has occurred in many parts of this manuscript. Therefore, if you decided to take action, please review the entire paper and rephrase where needed.
Authors: Good point. Corrections were made.
- Line 273: Change ‘Except’ to ‘except’
Authors: It was changed.
- Line 414-416: This sentence needs to be modified as your manuscript is a review not an original study. You may say ‘our analysis of the existing data revealed that a lower time of training …..
Authors: It has been modified. Thank you.

Reviewer 2 Report (New Reviewer)
This manuscript compares the effects of HiiT vs. no training on body fat reduction. Below are my main criticisms, later my specific comments will be listed.
Main points:
(1) There are already several meta analyses on this topic, where is the added value of this current work? The novelty of this research should definitely be emphasized. Why is prior research insufficient, what new insights are provided by this analysis?
(2) What is the relevance of these findings to practice? What are the consequences that need to be drawn from these results. It is not clear what new insights these results provide compared to existing studies.
(3) In this pairwise comparison, high intensity interval training was compared to no training at all. So it is not surprising that exercise (in this case HiiT) produces better effects in terms of body fat reduction. It is imperative to justify this comparison. If exercise were used for body fat reduction, yes would not be the option. Either to do high intensity endurance training or to do no training at all. Rather, options such as low-intensity endurance training would certainly be relevant. This should definitely be addressed.
Specific comments:
Introduction
- L93 Please add possible impacts of your results.
- L93 Furthermore, add some assumption about the results.
- L93 Please consider to use same paragraphs for the introduction, which would improve readability.
Methods
- L97 Was the screening & data extraction independently performed by to researcher?
Results
- L175 Please try to reduce redundancy between text and figures.
- L203 I have no access to table 3.
- L217 „aged 18 years and over“ Please give mean with SD
- L247 Please provide figures in highly quality
- L247 please discuss the high I squared values (in the discussion section). What does this mean for the interpretation of the results
- L305 Pleas double checke values from CHIn et al. 2020
-
Discussion
- L324 please do not only take significances into account. What about effect sizes. That should be kept in mind for the entire discussion.
- L331 When interpreting the results, not only the mean effect sizes should be considered, but also the associated variances.
Author Response
This manuscript compares the effects of HiiT vs. no training on body fat reduction. Below are my main criticisms, later my specific comments will be listed.
Authors: Thank you for the feedback and your time spent reviewing our work. We are pleased to hear you feel this work may have useful applications, and we appreciate your recommendations for further improvement.
Main points:
(1) There are already several meta analyses on this topic, where is the added value of this current work? The novelty of this research should definitely be emphasized. Why is prior research insufficient, what new insights are provided by this analysis?
Authors: We would like to kindly mention that this is the first systematic review and meta-analysis to assess the effects of High-Intensity Interval Training type (cycling vs. overground running vs. treadmill running) on fat mass (FM), body fat percentage (BF%), and fat-free mass (FFM). Numerous studies and systematic reviews (such as PMID: 31401727) have investigated the effects of HIIT on body composition in active or inactive individuals in the last two decades. Nevertheless, no systematic review has assessed the effect of the exercise type used. Considering exercise modality can influence the type and magnitude of health adaptations gained, we investigated whether the type of HIIT performed significantly alters body compositional changes. Findings from this systematic review and meta-analysis indicate advantageous body composition outcomes following HIIT (all modalities combined), including overall reductions in BF%, FM, and improved FFM. Specifically, improvements in BF% were enhanced following overground running HIIT compared to other exercise modalities. Another novel finding was that cycling HIIT was the most effective mode for significantly improving FFM.
(2) What is the relevance of these findings to practice? What are the consequences that need to be drawn from these results. It is not clear what new insights these results provide compared to existing studies.
Authors: To date, there is no consensus on which type of HIIT produces the greatest improvements in body composition. We expect that the new knowledge gained from this study will be a valuable resource for clinicians, practitioners, and researchers in the exercise and sports science field. It is important to understand what type of HIIT can lead to the most beneficial changes in body composition, as this will provide robust evidence for practitioners and coaches to develop effective and practical HIIT programs aimed at maximizing body compositional changes for health.
The following consequences can be drawn from the results of our systematic review and meta-analysis:
1-HIIT is an effective form of exercise for improving body composition, regardless of the mode of exercise.
2-Overground running may be more effective for reducing body fat percentage than treadmill running or cycling.
3-Cycling may be the most effective mode of HIIT for increasing FFM.
4-These findings suggest that individuals who want to reduce their body fat percentage may benefit more from overground running, while those who want to increase their FFM may benefit more from cycling.
Additions were made in lines 333-334.
(3) In this pairwise comparison, high intensity interval training was compared to no training at all. So it is not surprising that exercise (in this case HiiT) produces better effects in terms of body fat reduction. It is imperative to justify this comparison. If exercise were used for body fat reduction, yes would not be the option. Either to do high intensity endurance training or to do no training at all. Rather, options such as low-intensity endurance training would certainly be relevant. This should definitely be addressed.
Authors: We would like to kindly mention that we assessed the effects of High-Intensity Interval Training type (cycling vs. overground running vs. treadmill running) on fat mass (FM), body fat percentage (BF%), and fat-free mass (FFM). Consequently, evaluating low-intensity endurance training and how it compares to HIIT is not part of the aims of our study. Indeed, in our investigation, comparisons were made between different types of HIIT modes, and also compared to the non-exercise control group. The vast majority of systematic reviews and meta-analyses in our field include non-training controls as a comparison. Some of the benefits of using non-exercise control groups in scientific studies include:
1-Isolating the effects of the intervention: A non-exercise control group helps to isolate the effects of the intervention from other factors that may affect the outcome of the study. This allows researchers to better understand the specific effects of the intervention.
2-Reducing bias: A non-exercise control group can help to reduce bias in the study. Without a control group, it can be difficult to determine whether the observed effects are due to the intervention or other factors.
3-Providing a baseline for comparison: A non-exercise control group can provide a baseline for comparison to determine the magnitude of the effects of the intervention. This can help to determine whether the effects of the intervention are clinically meaningful.
4-Enhancing the generalizability of results: A non-exercise control group can help to enhance the generalizability of results by providing a comparison group that is similar to the intervention group in terms of age, sex, health status, and other relevant factors. This can help to ensure that the results are applicable to a wider population.
Specific comments:
Introduction
- L93 Please add possible impacts of your results.
Authors: Additions were made.
- L93 Furthermore, add some assumption about the results.
Authors: Additions were made.
- L93 Please consider to use same paragraphs for the introduction, which would improve readability.
Authors: It was included as the introduction.
Methods
- L97 Was the screening & data extraction independently performed by to researcher?
Authors: The first author performed screening and data extraction. This has been added to the main text.
Results
- L175 Please try to reduce redundancy between text and figures.
Authors: Corrected.
- L203 I have no access to table 3.
Authors: It was submitted in a separate file but the editorial office did not attach it for you. However, we will make sure it is now included.
- L217 „aged 18 years and over“ Please give mean with SD
Authors: The mean age was reported as an overall, L219.
- L247 Please provide figures in highly quality
Authors: Done.
- L247 please discuss the high I squared values (in the discussion section). What does this mean for the interpretation of the results.
Authors: The I2 index can be interpreted as the percentage of the total variability in a set of effect sizes due to true heterogeneity, that is, to between-studies variability. For example, a meta-analysis with I2 0 means that all variability in effect size estimates is due to sampling error within studies. This was already discussed in the limitation section.
- L305 Pleas double checke values from CHIn et al. 2020
Authors: We checked and it was correct.
Discussion
- L324 please do not only take significances into account. What about effect sizes. That should be kept in mind for the entire discussion.
Authors: Corrections were made.
- L331 When interpreting the results, not only the mean effect sizes should be considered, but also the associated variances.
Authors: This has now been considered. Thank you.

Reviewer 3 Report (New Reviewer)
General comments to the authors: Thank you for the opportunity to review your original manuscript. I am experienced with HIIT studies and well-versed in body composition assessment, but I am unfamiliar with the statistical analyses you used, so do keep that in mind as you read my comments. Given the title and purpose of the manuscript, this reviewer was expecting to see a level of focus on the body composition assessment methods and results determination similar to that given the subgroup analyses of the HIIT protocol heterogeneity.
Specific comments: Section 2.2: for your inclusion criteria, were you evaluating absolute or relative changes in FM, FFM, and total body mass, or did you just take the value reported and use that? What scrutiny was given in regard to the body composition aspects such as: pre-test guidelines, dietary control, participant hydration status, use of measured vs estimated residual lung volume (hydrostatic weighing)/thoracic gas volume (ADP), conversion formulae selected, etc.?
Sections 3.3 and 3.4: It appears that you relied on the two-component model concept in which total body mass = FM+FFM. When discussing significant reductions in BF%, what were the corresponding changes in FFM and TBM? Body water, especially for women, can fluctuate by 1-3 pounds from one day to the next; that will increase FFM and mathematically indicate that there has been a loss in FM and %BF.
Section 4: You mention that there is a significant pooled reduction in %BF of 1.53%; this reviewer suggests you consider mentioning if it is also clinically or practically significant given that it is within the error range of several of the methods used to estimate body composition? You mention in the Discussion section that there could be a limitation based on the magnitude of the estimation error within the respective method(s) used in the various studies selected for inclusion. This reviewer believes it would be of benefit to create a table showing, by included study, the device used for collection of FM, FFM, and %BF as well the resulting SEE, SEM and total error.
Table 2. This table would be easier to ready and understand if it were in landscape format.
Reference list: This is the first time this reviewer has seen the reference list be right-justified. Please check the author guidelines for the formatting of the reference list for this as well as for use of capitalization in the article titles/journal names and abbreviation of journal names.
Author Response
General comments to the authors: Thank you for the opportunity to review your original manuscript. I am experienced with HIIT studies and well-versed in body composition assessment, but I am unfamiliar with the statistical analyses you used, so do keep that in mind as you read my comments. Given the title and purpose of the manuscript, this reviewer was expecting to see a level of focus on the body composition assessment methods and results determination similar to that given the subgroup analyses of the HIIT protocol heterogeneity.
Authors: Thank you for the feedback and your time spent reviewing our work. We are pleased to hear you feel this work may have useful applications, and we appreciate your recommendations for further improvement. The assessment of the body composition using various equipment is shown in table 3. We also acknowledged this in the limitation section.
Specific comments: Section 2.2: for your inclusion criteria, were you evaluating absolute or relative changes in FM, FFM, and total body mass, or did you just take the value reported and use that? What scrutiny was given in regard to the body composition aspects such as: pre-test guidelines, dietary control, participant hydration status, use of measured vs estimated residual lung volume (hydrostatic weighing)/thoracic gas volume (ADP), conversion formulae selected, etc.?
Authors: Absolute changes are evaluated. You brought up a good point. Some studies did not report the pre-test guidelines or hydration control which may be due to various reasons (word limitation, etc.,). We acknowledged this important point in the limitations.
Sections 3.3 and 3.4: It appears that you relied on the two-component model concept in which total body mass = FM+FFM. When discussing significant reductions in BF%, what were the corresponding changes in FFM and TBM? Body water, especially for women, can fluctuate by 1-3 pounds from one day to the next; that will increase FFM and mathematically indicate that there has been a loss in FM and %BF.
Authors: We appreciate these excellent points. However, as this is a systematic review and meta-analysis, and since the included studies do not primarily highlight the points mentioned, we cannot accurately consider them. This is also applicable to all published meta-analytic studies in the field, not just ours. Additionally, we would like to note that out of the studies included, 35 evaluated BF%, while only 9 evaluated FFM.
Section 4: You mention that there is a significant pooled reduction in %BF of 1.53%; this reviewer suggests you consider mentioning if it is also clinically or practically significant given that it is within the error range of several of the methods used to estimate body composition? You mention in the Discussion section that there could be a limitation based on the magnitude of the estimation error within the respective method(s) used in the various studies selected for inclusion. This reviewer believes it would be of benefit to create a table showing, by included study, the device used for collection of FM, FFM, and %BF as well the resulting SEE, SEM and total error.
Authors: Thank you for raising these points. There are clinical benefits for the magnitude in BF% changes we report. For instance, a prior study indicated that a 1% reduction in body fat was associated with a 3% reduction in the risk of developing type 2 diabetes [1]. Additions were made in lines 263-266.
Regarding the table point, Table 3 was submitted separately and was not sent out to reviewers; however, we have now resubmitted it so the reviewers can see all the characteristics of the included studies.
Reference
1- Hinnouho, G. M., Czernichow, S., Dugravot, A., Nabi, H., Brunner, E. J., Kivimaki, M., & Singh-Manoux, A. (2015). Metabolically healthy obesity and the risk of cardiovascular disease and type 2 diabetes: the Whitehall II cohort study. European heart journal, 36(9), 551-559.
Table 2. This table would be easier to ready and understand if it were in landscape format.
Authors: Corrections were made.
Reference list: This is the first time this reviewer has seen the reference list be right-justified. Please check the author guidelines for the formatting of the reference list for this as well as for use of capitalization in the article titles/journal names and abbreviation of journal names.
Authors: Corrections were made. Thank you.

Round 2
Reviewer 2 Report (New Reviewer)
All the points I mentioned were dealt with. However, my main points were only "disarmed" and not solved. Accordingly, I do not expect a high reach of this paper. Overall, however, it still meets the scientific standards.
Reviewer 3 Report (New Reviewer)
Very nice job with Table 3 and additions re: limitations for body composition evaluation.
This manuscript is a resubmission of an earlier submission. The following is a list of the peer review reports and author responses from that submission.
Round 1
Reviewer 1 Report
Dear authors
The paper is well written, however, in my opinion there are too many types of training programs being compared, with too many outcomes. The different HIT protocols differ much in bouts, intensity and treatment duration, and the difference in effects between 3-week duration studies and 15-week worries me. A finer cut-off than </> 8 wks could be interesting. Are e.g. most of the cycling protocols of short duration? And are the training regimes in the studies supervised? I miss a discussion of the clinical relevance of the relative small differences in FM reported.
The figure quality needs improvement, and the title of the tables more explanatory without abbreviations. The figure/table abbreviations are not explained in the legends. In opinion, there are too many tables.
I hope you can use my comments to improve the paper.
Author Response
Dear authors
The paper is well written, however, in my opinion there are too many types of training programs being compared, with too many outcomes. The different HIT protocols differ much in bouts, intensity and treatment duration, and the difference in effects between 3-week duration studies and 15-week worries me. A finer cut-off than </> 8 wks could be interesting. Are e.g. most of the cycling protocols of short duration? And are the training regimes in the studies supervised? I miss a discussion of the clinical relevance of the relative small differences in FM reported.
Authors: We wish to thank the reviewer for their insightful comments and improving the quality of our work. We agree with the reviewer regarding the ambiguity and heterogeneity of the HIIT protocols included in our work and have therefore now included further subgroup analysis (e.g., frequency, time of training [period], rest time, passive or active) in Table 4 in our amended manuscript to increase the clarity of our results and subsequent discussion.
The figure quality needs improvement, and the title of the tables more explanatory without abbreviations. The figure/table abbreviations are not explained in the legends. In opinion, there are too many tables.
Authors: We thank the reviewer for this suggestion and have now improved the quality of our figures as well as included further detail and expanded abbreviations in our Table legends..
I hope you can use my comments to improve the paper.
Authors: All your comments were outstanding and we appreciate the time you devoted to precisely read our paper.

Reviewer 2 Report
This is an overall very interesting and well written manuscript about the effect of HIIT on body composition with very few comments.
Major Comment:
Some studies included in this meta-analysis are from obese, overweight and some from normal weight subjects. One would expected that obese subjects would lose more fat than overweight and overweight more than norma weight. It would be nice to present forest plots for obese, overweight and normal weight subjects separately.
Minor comment:
Forest plots should be provided in higher resolution images.
Author Response
This is an overall very interesting and well written manuscript about the effect of HIIT on body composition with very few comments.
Major Comment:
Some studies included in this meta-analysis are from obese, overweight and some from normal weight subjects. One would expected that obese subjects would lose more fat than overweight and overweight more than normal weight. It would be nice to present forest plots for obese, overweight and normal weight subjects separately.
Authors: We thank the reviewer for raising this important point. However, we kindly mention that if we present forest plots for the mentioned subjects separately, it is likely to increase confusion for readers since currently we have them based off cycling, overground running and treadmill running. If we present them separately, the number of plots will be nine for FM, BF%, and FFM, which means we will have a total of 27 plots which is not feasible
Minor comment:
Forest plots should be provided in higher resolution images.
Authors: We thank the reviewer for raising this point and have now amended accordingly.
